# Reproductive compatibility in *Capsicum* is not necessarily reflected in genetic or phenotypic similarity between species complexes

Catherine Parry[1], Yen-Wei Wang[2], Shih-wen Lin[2], Derek W. Barchenger[2]*

1 Department of Biology and Biochemistry, University of Bath, Claverton Down, Bath, United Kingdom,
2 World Vegetable Center, Shanhua, Tainan, 74151, Taiwan

* derek.barchenger@worldveg.org

**Data Availability Statement:** The data used in this study are available in the World Vegetable Center repository, HARVEST (DOI: 10.22001/wvc.73914).

## Abstract

Wild relatives of domesticated *Capsicum* represent substantial genetic diversity and thus sources of traits of potential interest. Furthermore, the hybridization compatibility between members of *Capsicum* species complexes remains unresolved. Improving our understanding of the relationship between *Capsicum* species relatedness and their ability to form hybrids is a highly pertinent issue. Through the development of novel interspecific hybrids in this study, we demonstrate interspecies compatibility is not necessarily reflected in relatedness according to established *Capsicum* genepool complexes. Based on a phylogeny constructed by genotyping using simple sequence repeat (SSR) markers and with a portion of the *waxy* locus, and through principal component analysis (PCA) of phenotypic data, we clarify the relationships among wild and domesticated *Capsicum* species. Together, the phylogeny and hybridization studies provide evidence for the misidentification of a number of species from the World Vegetable Center genebank included in this study. The World Vegetable Center holds the largest collection of *Capsicum* genetic material globally, therefore this may reflect a wider issue in the misidentification of *Capsicum* wild relatives. The findings presented here provide insight into an apparent disconnect between compatibility and relatedness in the *Capsicum* genus, which will be valuable in identifying candidates for future breeding programs.

## Introduction

The genus *Capsicum* (n = 12 or 13) is comprised of about 35 diploid species including five domesticated species: *C. annuum* L., *C. baccatum* L., *C. chinense* Jacq., *C. frutescens* L., and *C. pubescens* Ruiz & Pav. [1]. All members of the genus originate in the Americas; however, the crop is produced worldwide with the majority of production occurring in Asia [2]. The genetic and phenotypic diversity across the genus is significant, and thus represents a valuable resource for crop improvement [2]. The primary limitations to improving productivity and quality of *Capsicum* are abiotic and biotic stresses, many of which lack sources of host tolerance or resistance [3]. Furthermore, as a widely consumed crop with cultural and culinary

**Funding:** Funding for this research was provided by the Ministry of Science and Technology (MOST) of Taiwan (Project ID:107-2311-B-125 -001 -MY3) and long-term strategic donors to the World Vegetable Center: Taiwan, UK aid from the UK government, United States Agency for International Development (USAID), Australian Centre for International Agricultural Research (ACIAR), Germany, Thailand, Philippines, Korea, and Japan.

**Competing interests:** The authors have declared that no competing interests exist.

value across global cuisines, there is high demand for *Capsicum* [4]. There is therefore significant incentive to overcome challenges to cultivation, one means of doing so being the introgression of resistance to the various stresses that limit production of *Capsicum* species.

Understanding interspecies compatibility and identifying barriers to hybridization is essential to the design of introgression breeding programs. *Capsicum* species are divided among 11 clades [4,5] and grouped into three complexes—Annuum, Baccatum and Pubescens—based on their relative reproductive compatibility [6–8]. There is understood to be relatively low reproductive compatibility between species complexes [9], while unknown mechanisms of unilateral incompatibility have previously been demonstrated [10]. Barriers to hybridization may include failure of the pollen grain to germinate or the pollen tube to develop, or may be post-zygotic: embryo death or inviability, such as that caused by untolerated aneuploidy [11,12]. The pre- and post-zygotic barriers to hybridization between genetic complexes in *Capsicum* remains largely unresolved [13], however, a number of cross-complex hybridizations have been achieved [13–18]. This suggests isolation between complexes is not absolute, and there is therefore potential for introgression breeding, or design of genetic bridge strategies in order to best exploit this genetic variation.

In contrast to other Solanaceae crops, including tomato (*Solanum lycopersicum* L.) [19], potato (*S. tuberosum* L.) [20] and to a lesser extent eggplant (*S. melongena* L.) [21], introgression breeding using wild species has been relatively underutilized in *Capsicum*; [22]. The wild progenitor, *C. annuum* L. var. *glabriusculum* (Dunal) Heiser & Pickersgill is a potential source of disease resistance, with reported resistance to Beet curly top virus (BCTV: *Curtovirus*) [23,24]. Members of the wild species *C. chacoense* (Hunz.) and *C. rhomboideum* (Dunal) Kuntze have been identified as being resistant to powdery mildew (*Leveillula taurica*) [25]. Recently, an accession of *C. galapagoense* Hunz. has been proposed to be a potential source of resistance to the insect pest, whitefly, based on trichome density and type (M. Rhaka, pers. comm.). However, despite extensive hybridization no successful progeny have so far been developed [26]. These results are surprising because *C. galapagoense* has been reported as part of the *C. annuum* clade, and readily hybridize with *C. annuum* accessions [5,7]. One reason for unsuccessful hybridization attempts may be misidentification; several genebanks have incorrectly reported accessions identified as *C. galapagoense* which are, in fact, *C. frutescens* (P. W. Bosland, pers. comm.). Such misidentification presents a challenge to utilizing knowledge of the relatedness of *Capsicum* species and their ability to hybridize. Although the genetic diversity and variation within wild populations of *Capsicum* has been studied [5,27–31], the pool of phenotypic data for wild *Capsicum* species remains limited [2]. There also remains a lack of access to publicly available germplasm representing the diversity of wild *Capsicum* [1]. There is therefore an immediate need to better understand the role of wild *Capsicum* species in future breeding programs.

The objectives of this study were to elucidate the relationship between interspecies compatibility and relatedness through extensive interspecific hybridization and the construction of a phylogeny. We aimed to clarify the relationships among the wild and domesticated *Capsicum* species included in the study, and confirm the identities of several World Vegetable Center genebank accessions.

## Materials and methods

Thirty-eight accessions of 15 species of *Capsicum* were chosen for this experiment (Table 1). Most of the accessions in our experiment have been previously karyotyped and have 12 chromosomes, with the exceptions of *C. eshbaughii* Barboza (n = unknown), *C. minutifolium* (Rusby) Hunz. (n = unknown) and *C. rhomboideum* (n = 13). The accessions were provided to

**Table 1.** *Capsicum* accessions included in this study.

| Species | Accession | Cultivar or other name | Source[a] |
|---|---|---|---|
| *Capsicum annuum* L. | AVPP9905 | Susan's Joy | WorldVeg |
| | Criollo de Morelos 334 (CM334) | PBC 1867 | NMSU |
| | California Wonder | PBC 196 | WorldVeg |
| | PBC 1799 | Bird Pepper | WorldVeg |
| | VI059328 | PBC 142 | WorldVeg |
| | VI029657 | | WorldVeg |
| *Capsicum annuum* L. var. *glabriusculum* (Dunal) Heiser & Pickersgill | PI 574547 | Chile que mira p'arriba, PBC 1969 | USDA-ARS |
| | PI 674459 | BG2816 selection 16–1, PBC 1970 | USDA-ARS |
| *Capsicum baccatum* L. | VI012528 | | WorldVeg |
| | VI014924 | Aje | WorldVeg |
| | PBC 80 | | WorldVeg |
| | PBC 81 | Jin's Delight | WorldVeg |
| | VI012478 | | WorldVeg |
| *Capsicum cardenasii* Heiser & P.G. Sm. | NMCA90030 | PBC 1987 | NMSU |
| | NMCA90035 | PBC 1989 | NMSU |
| *Capsicum chacoense* Hunz. | VI012574 | PBC 814 | WorldVeg |
| | VI012900 | | WorldVeg |
| *Capsicum chinense* Jacq. | PI 159236 | 30040 | USDA-ARS |
| | PI 152225 | Miscucho Colorado | USDA-ARS |
| | VI012668 | PBC 306 | WorldVeg |
| | VI029446 | | WorldVeg |
| | PBC 1793 | Scotch Bonnet Pepper | WorldVeg |
| *Capsicum eximium* Hunz. | VI013161 | | WorldVeg |
| | VI012964 | | WorldVeg |
| *Capsicum eshbaughii* Barboza | NMCA90006 | PBC 1990 | NMSU |
| *Capsicum flexuosum* Sendtn. | NMCA50030 | PBC 1991 | NMSU |
| | NMCA50034 | PBC 1992 | NMSU |
| *Capsicum frutescens × chinense* | PBC 1820 | Bhut Jolokia | WorldVeg |
| *Capsicum frutescens* L. | PBC 556 | MC-003 | WorldVeg |
| *Capsicum galapagoense* Hunz. | NMCA50026 | PBC 1892 | NMSU |
| | VI051011 | | WorldVeg |
| *Capsicum minutifolium* (Rusby) Hunz. | NMCA50053 | PBC 1993 | NMSU |
| *Capsicum praetermissium* Heiser & P.G. Sm. | NMCA90027 | PBC 1887 | NMSU |
| | VI029696 | | WorldVeg |
| | VI029697 | | WorldVeg |
| *Capsicum rhomboideum* (Dunal) Kuntze | NMCA50017 | PBC 1995 | NMSU |
| | NMCA50064 | PBC 1996 | NMSU |
| *Capsicum tovarii* Eshbaugh et al. | VI051012 | | WorldVeg |

[a]Source organization abbreviations: WorldVeg, The World Vegetable Center; NMSU, New Mexico State University; USDA-ARS, United States Department of Agriculture–Agricultural Research Services.

the World Vegetable Center, having been collected from diverse locations and deposited into collections at either the World Vegetable Center Genebank, the World Vegetable Center Pepper Breeding Collection in Tainan, Taiwan, the United States Department of Agriculture—Agriculture Research Service National Plant Germplasm System, or the Chile Pepper Institute, New Mexico State University, Las Cruces, NM USA. Of each accession, two biological

replications were used wherever possible for phenotyping and genotyping, although due to poor germination, four accessions (NMCA50034, PBC 556, PBC 1892, NMCA50064) did not have a biological replicate.

All experiments were conducted at the World Vegetable Center, Shanhua, Tainan, Taiwan (lat. 23.1˚N; long. 120.3˚E; elevation 12 m). Prior to sowing, all seed was treated with trisodium phosphate (TSP) and hydrochloric acid (HCl) following the methods of Kenyon et al. [32], which has been observed to reduce germination rates. Seeds were sown into 72-cell plastic trays of sterilized peat moss. Trays were placed in a climate-controlled greenhouse for germination at 28 ± 3˚C with a 12-hour photoperiod and ≈95% relative humidity. At the 4–6 true leaf stage, the seedlings were transplanted into pots and moved to a greenhouse without climate control. Plants were irrigated twice daily and regularly fertilized with Nitrophoska (Incitec Pivot Fertilisers, Victoria, Australia) during the experimental period.

The accessions were morphologically characterized according to the Descriptors of *Capsicum* Manual [33] for the following characteristics: mature leaf length, mature leaf width at widest point, leaf color, density (if present) of leaf pubescence, leaf shape, lamina margin, stem color, stem shape, density (if present) of stem pubescence, nodal anthocyanin color, node length, anther color, anther length, filament length, corolla color, corolla spot color, corolla shape, corolla length, stigma exsertion, flower position, tillering, leaf density, fruit length, fruit width, fruit pedicel length, neck at base of fruit. Quantitative traits were the mean of 10 values measured across replicates. Qualitative traits were scored according to the IPGRI Descriptors of *Capsicum* manual [33] based on observations of both plant replicates. Accessions with incomplete data were excluded from analysis of phenotypic data. To identify trends in traits between species, the quantitative traits were used for principal component analysis (PCA) using the R packages, 'factoextra' [34] and 'ggfortify' [35] for PCA analysis with scaling. The scores of qualitative traits were analyzed using an unweighted pair group method with arithmetic mean (UPGMA) hierarchical cluster analysis. Bootstrap resampling was applied to clustering with 1,000 iterations.

Reciprocal hybridizations were attempted among all combinations of accessions throughout the experimental period. Ability to hybridize in reciprocal was used to confirm previous reports of relatedness and ability to hybridize species across clades and complexes. The fruits of successful hybridizations were collected upon ripening. Within three days of harvest, the seeds were extracted from the fruits and dried for at least 1 week. Five seeds each of 112 crosses of interest were sown into 72-cell plastic trays containing sterilized peat moss. The trays were placed in a greenhouse without climate control and irrigated twice daily and observed daily for 12 weeks to assess germination. A chord diagram was produced in R using the package 'circlize' [36] to visualize successful crosses for which seed was obtained. A heat map was produced in R using the package '[37] to visualize the percentage of seeds germinated after 12 weeks.

For genotyping, DNA was isolated from young, actively growing leaves from plants of each accession using the modified cetrimonium bromide (CTAB) extraction method [38]. Using 27 Simple Sequence Repeat (SSR) markers, DNA was amplified by PCR, for which each well of a 96-well microtiter plate contained 2 μl of template DNA, 0.4 μl of primer (0.2 μl each forward and reverse), 0.1 μl of AmpliTaq Gold DNA polymerase, 0.4 μl of deoxyribonucleotides, 1.5 μl of 10× PCR Buffer II Gold buffer (Thermo Fisher Scientific, Waltham, MA, USA), and sterile water to a final volume of 15 μl. The reactions were carried out in a thermal cycler (Single Block Alpha Unit, DNA Engine®, Bio-Rad Laboratories, Berkeley, CA, USA) with an annealing temperature of 55˚C. The electrophoresis of amplified products was performed on 6% acrylamide gels at 160 Volts for 30 minutes (Thermo Electron Electrophoresis EC250-90, Thermo Fisher Scientific). The results were visualized under UV light using UVITEC Imaging

Systems (Cleaver Scientific, Warwickshire, UK) following staining with ethidium bromide. Electrophoresis was repeated whenever the clarity of the bands or their exact size was uncertain.

Gels were scored for each primer pair using a binary method: each accession was scored for presence (1) or absence (0) of amplicons of each size. The data were processed in R using the packages, 'proxy' and 'shipunov' [39] to produce a dendrogram with bootstrapping for the assessment of the relatedness between the individual accessions. A distance matrix was produced using the Dice index, and an unweighted pair group method with arithmetic mean (UPGMA) hierarchical cluster analysis was carried out. Bootstrap resampling was applied to clustering with 1,000 iterations.

Further molecular analysis to clarify the identification of some accessions included the study of the *waxy* gene region of six accessions, VI051012 (*C. tovarii*); VI051011 (*C. galapagoense*, potentially *C. annuum*); VI012574 (*C. chacoense*, potentially *C. annuum*); PBC 1892 (*C. galapagoense*); VI013161 (*C. eximium* Hunz); and PBC 556 (*C. frutescens*), using the primer pair, 860F and 2R [5]. The chosen accessions were those expected to need clarification due to possible misidentification, based on molecular and morphological data. The *waxy* region was amplified by PCR as before, with an annealing temperature of 60°C. The quality of the products were evaluated by running on a 2% agarose gel with EtB'out' (Yeastern Biotech Co. Ltd., Taipei, Taiwan) at 100 Volts for 50 minutes, then visualized using a Microtek Bio- 1000F gel imager (Microtek International Inc., Hsinchu, Taiwan). The PCR products were sequenced by Genomics Biotechnology Co., Ltd. (New Taipei City, Taiwan) by the Sanger sequencing method. Low-quality nucleotides were manually removed throughout the resulting sequence, including approximately the first and last 60 nucleotides. The sequences were aligned using NCBI nucleotide BLAST [40] and a consensus sequence constructed using the CAP contig assembly program from BioEdit [41]. The sequences, including that of the publicly available *S. lycopersicum* GBSS sequence (gene ID: 101259777) as the outgroup, and the *waxy* sequences of 15 *Capsicum* species deposited in NCBI (accession numbers: KP747352.1, KP747351.1, KP747358.1, KP747354.1, KP747353.1, KP747360.1, KP747310.1, KP747309.1, KP747359.1, KP747314.1, KP747306.1, KP747357.1, KP747311.1, KP747320.1, KP747361.1) (National Center for Biotechnology Information (NCBI) [42], were aligned using multiple sequence alignment tool, Clustal MAFFT [43]. The resulting dendrogram was visualized using Interactive Tree of Life (iTOL) version 5.7 [44].

## Results

To clarify the phylogeny of the wild and domesticated *Capsicum* species in the sample, UPGMA clustering was applied to the genetic variation captured by the SSR molecular markers. The *C. baccatum* and *C. chinense* group accessions were distinct from the *C. annuum* group, with 76% bootstrap support (Fig 1). The *C. baccatum* accessions made up a significant group, being closely clustered with the *C. praetermissium* Heiser & P.G. Sm. accessions. This grouping was adjacent to a large group comprised of closely clustered *C. chinense* accessions with the *C. galapagoense* accession PBC 1892, as well as *C. eshbaughii*, *C. eximium* and *C. frutescens*, separated from the *C. baccatum* group with a relatively low confidence interval. Within this grouping, *C. chinense* accession, PBC 1820, was distinct from its counterparts, with 92% bootstrap support. Furthermore, the *C. chinense* species accessions were relatively separate from the accessions at the periphery of this grouping, the *C. galapagoense* accession PCB 1892, the *C. frutescens* accession PBC 556, and the *C. eximium* accession VI013161. The grouping of the *C. eximium* accession VI013161 with *C. frutescens* was similar in clustering from the *waxy* gene sequence (Fig 2). These accessions were thus more similar to each other than they were

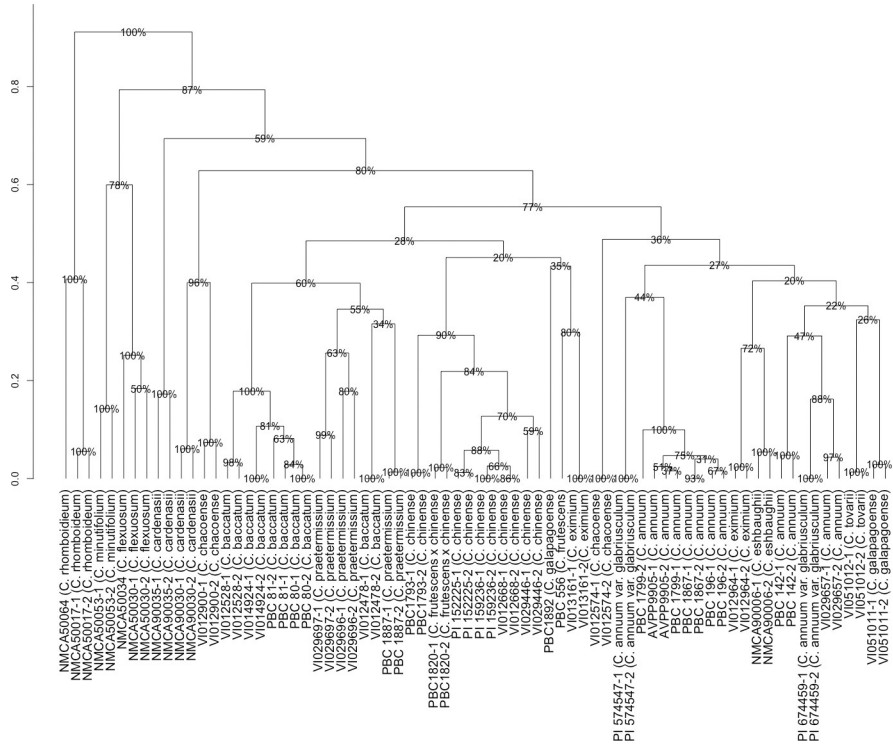

**Fig 1. Unweighted pair group method (UPGMA) clustering of *Capsicum* species according to simple sequence repeat (SSR) markers.** 'Height' represents dissimilarity, derived from 'dice method'. Bootstrap resampling applied to clusters, represented as percent confidence interval. Numbers following the hyphen indicate replicates.

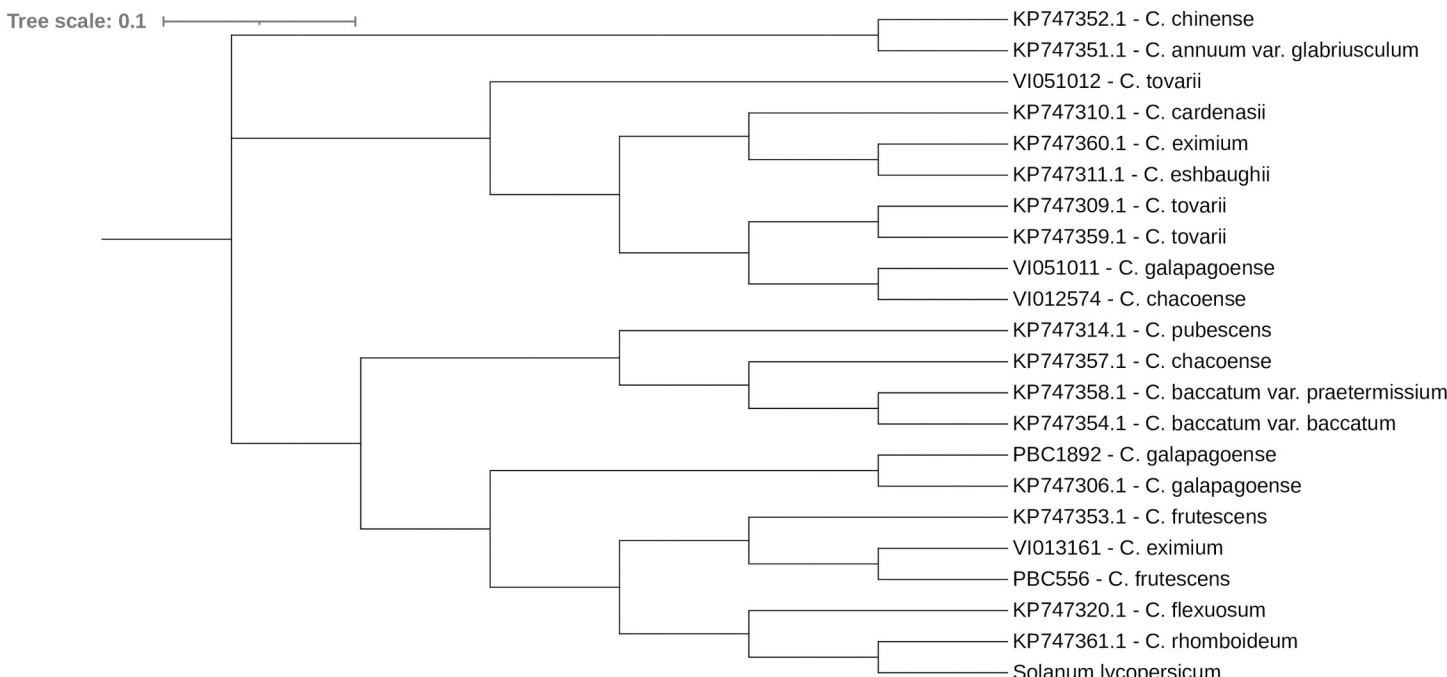

**Fig 2. Clustering of *Capsicum* species according to their *waxy* gene sequences.** Sequences of accessions that begin with "KP" were obtained from NCBI, while those that begin with "PBC" or "VI" were from this experiment. The *waxy* sequence (gene ID: 101259777) of tomato (*Solanum lycopersicum*) was used as the root of the tree.

similar to the *C. galapagoense* accession PBC 1892, and this was a distinct grouping from other sequenced accessions.

To provide further evidence of this phylogeny, we analyzed the *waxy* gene sequence of a sample of accessions in this study (VI051012, VI051011, VI012574, VI013161, and PBC 1892), that of a range of *Capsicum* species, and *S. lycopersicum*, available publicly (Fig 2). In this analysis, *C. baccatum* accessions were similarly clustered closely with *C. pubescens*, and with *C. chacoense*.

To better understand the crossing relationship between species, reciprocal hybridizations were performed between each combination of accessions (Fig 3). Members of *C. baccatum* and *C. praetermissium* hybridized as either the female or male parent with at least one accession of each other species with the exception of *C. rhomboideum* (Fig 3). However, of the sample of seeds selected for sowing, only the cross between VI014924 and PBC 1969 germinated (Fig 4). Hybridizations were not achieved between *C. galapagoense* as either parent with accessions of *C. tovarii*, *C. flexuosum*, *C. minutiflorium*, *C. cardenasii*, *C. eshbaughii*, and *C. rhomboideum* species. *Capsicum eshbaughii* hybridized more readily as the female parent, but failed to hybridize in either direction with accessions of *C. eximium*, *C. frutescens*, *C. galapagoense*, *C. tovarii*, *C. flexuosum*, and *C. rhomboideum*. The majority of *C. frutescens* hybrids were achieved with *C. annuum* accessions, but successful hybridizations were found across a broad species range. Of the sample of seeds sown, 80% of the PBC 556 × PBC 1970 cross seeds germinated (Fig 4).

The *C. annuum* group, which was adjacent to *C. baccatum*, consisted of the closely clustered *C. annuum* species accessions: PBC 1799, AVPP9905, PBC 1899, PBC 1867, and PBC 196, along with *C. annuum* var. *glabriusculum* PI 574547, and *C. chacoense* VI012574 (Fig 1). Neighboring this group was a cluster comprised of the *C. eximium* accession VI012964, the *C. eshbaughii* accession NMCA90006, the *C. annuum* accessions PBC 142 and VI029657, the *C. annuum* var. *glabriusculum* accession PI 674459, the *C. tovarii* Eshbaugh et al. accession VI051012, and the *C. galapagoense* accession VI051011. Based on clustering of the *waxy* gene sequence, we found *C. tovarii* accession VI051012 to be clustered broadly with *C. chacoense*, *C. galapagoensis*, *C. eximium* and *C. eshbaughii*, and with two other *C. tovarii* accessions (Fig 2). The *C. galapagoense* accession VI051011 was distinct from its counterpart, PBC 1892, and another *C. galapagensis* accession, KP747306.1 (Fig 2).

Accessions of *C. annuum* hybridized in both directions with one or more accessions of all species except *C. rhomboideum* and *C. minutifolium* (Fig 3). Thirteen of the hybrids sown germinated well (Fig 4). *Capsicum annuum* var. *glabriusculum* hybridized in either direction with at least one accession of every species except *C. rhomboideum*, and 7 out of 10 of those sown germinated (Fig 4). Accessions of *C. chacoense* also hybridized broadly, but not with *C. frutescens* × *chinense*, *C. minutifolium* or *C. rhomboideum*, and seven of the 25 hybrids sown germinated. More than one cross was achieved between *C. tovarii* and an accession of every species except *C. eshbaughii*, *C. eximium*, *C. galapagoense*, *C. minutifolium* and *C. rhomboideum*. Of these crosses, VI012574 × VI051012 and NMCA90030 × VI051012 germinated with 40% and 100% efficiency, respectively.

With 80% confidence interval, *C. chacoense* accession VI012900 and *C. cardenasii* Heiser & P.G. Sm. accession NMCA90030 were clustered separately from the *C. baccatum* and *C. annuum* groups (Fig 1). NMCA90035 clustered distinctly from its *C. cardenasii* counterpart, with bootstrap support of 57%. The *C. flexuosum* Sendtn. accessions NMCA50034 and NMCA50030 clustered closely together, with high bootstrap support (99%); adjacent was the *C. minutifolium* accession NMCA50053, and in a separate cluster, the *C. rhomboideium* accession NMCA50064, which was the most distinct grouping, separated from its neighbors with 100% confidence. This was supported by *waxy* sequences which showed *C. rhomboideum* to be

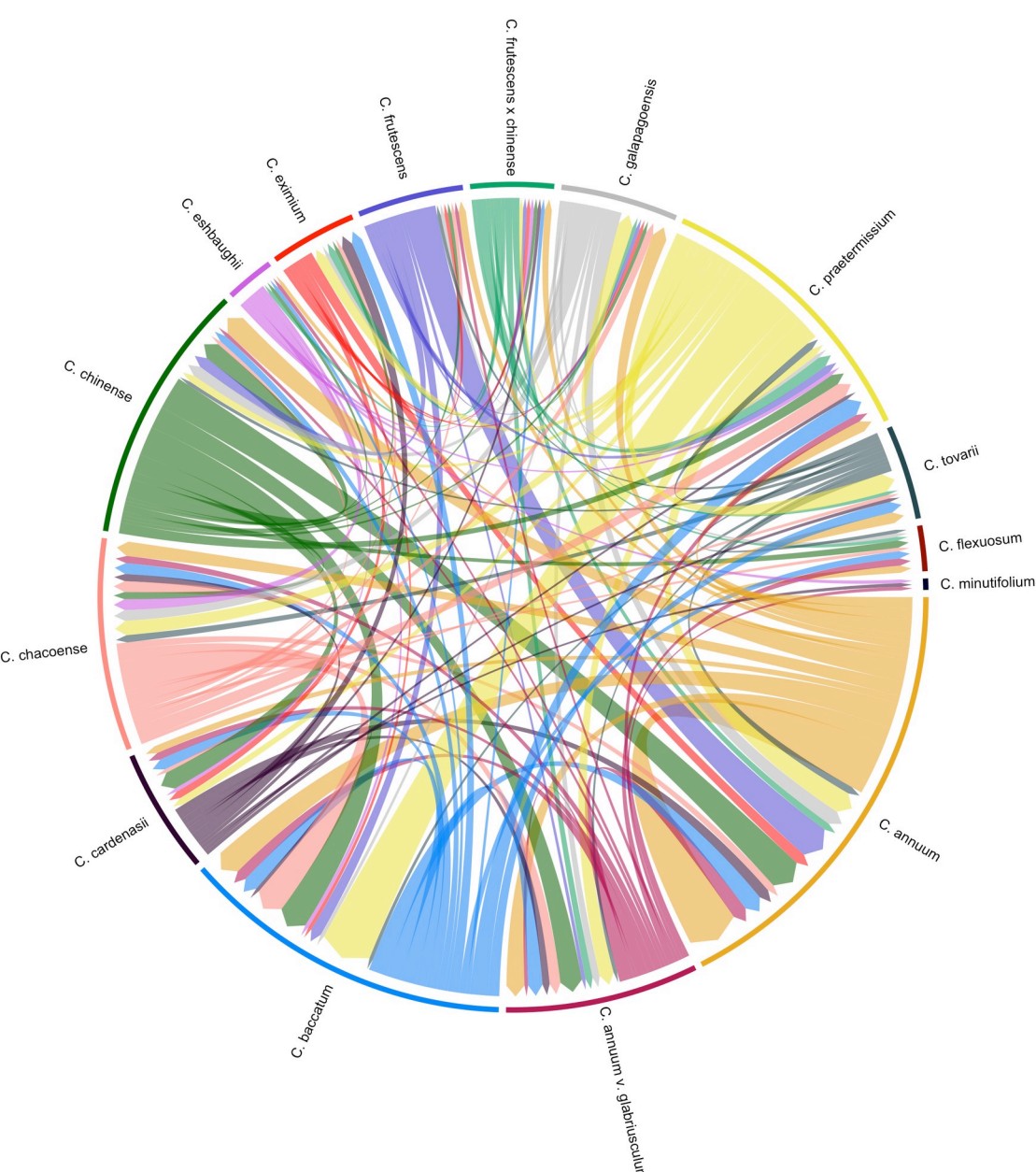

**Fig 3. Reciprocal hybridizations achieved between accessions of *Capsicum* species.** Direction of arrow represents successful hybridizations in the male-female direction from which fruit was harvested.

most similar to the outgroup, *S. lycopersicum*, and more broadly clustered with *C. flexuosum* (Fig 2).

*Capsicum cardenasii* species hybridized with every species except *C. frutescens*, *C. flexuosum*, *C. galapagoense* and *C. rhomboideum* (Fig 3), and the five of the nine hybrids sown germinated well (Fig 4). No hybrids were achieved with *C. flexuosum* or *C. minutifolium* as female parents, however *C. flexuosum* hybridized as the male parent with *C. chacoense*, *C. annuum* var. *glabriusculum*, *C. baccatum*, *C. tovarii*, *C. annuum* and *C. frutescens*, while *C. minutifolium* hybridized with *C. cardenasii*, *C. eshbaughii*, and *C. annuum* var. *glabriusculum*, and of the

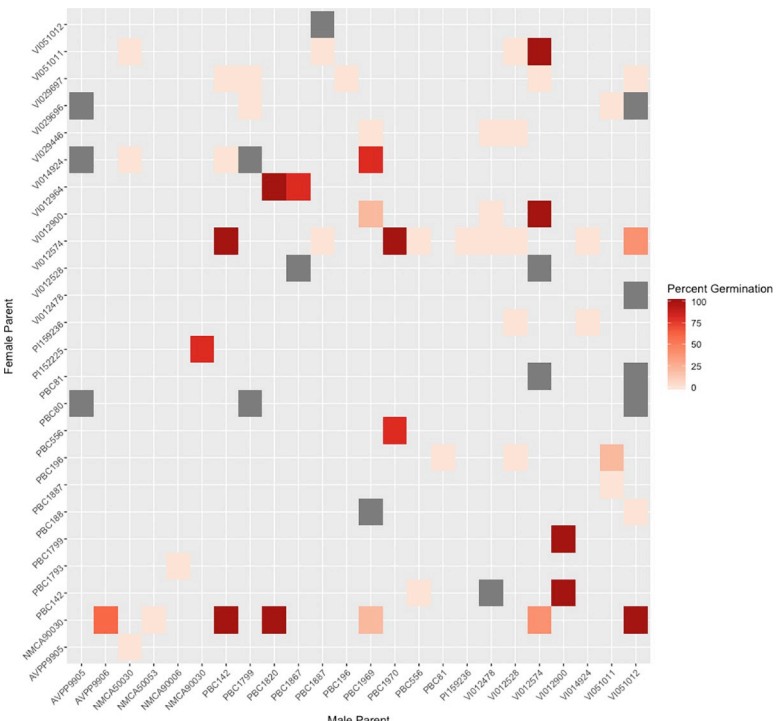

**Fig 4. Percent germination of selected hybrid seeds 12 weeks after sowing.** Grey indicates unviable seeds.

crosses sown, only VI012574 × PBC 124 germinated (Fig 4). No successful hybrids were achieved with *C. rhomboideum* in either direction (Fig 3).

We applied principal component analysis to the quantitative phenotypic data collected to understand phenotype across *Capsicum* species (Fig 5). The first two components account for

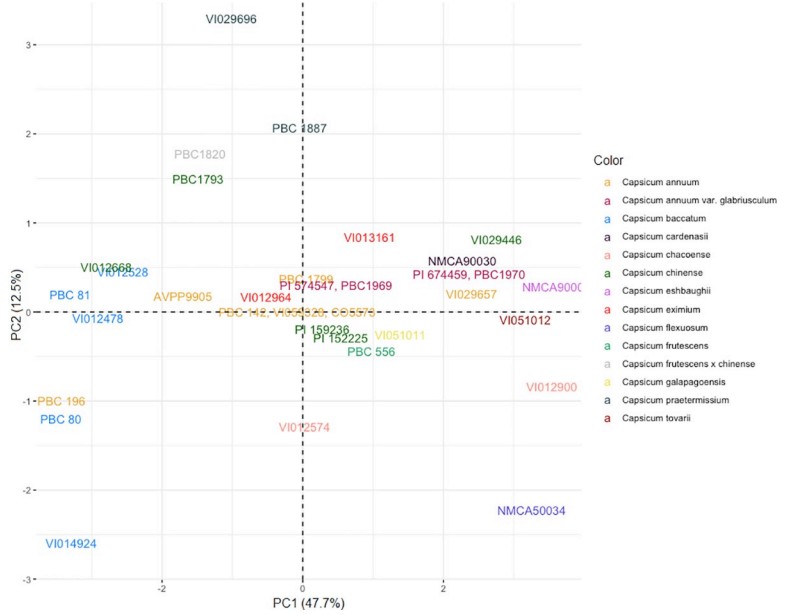

**Fig 5. First two principal components of accessions in the wild and domesticated *Capsicum* species based on the quantitative phenotypic data.**

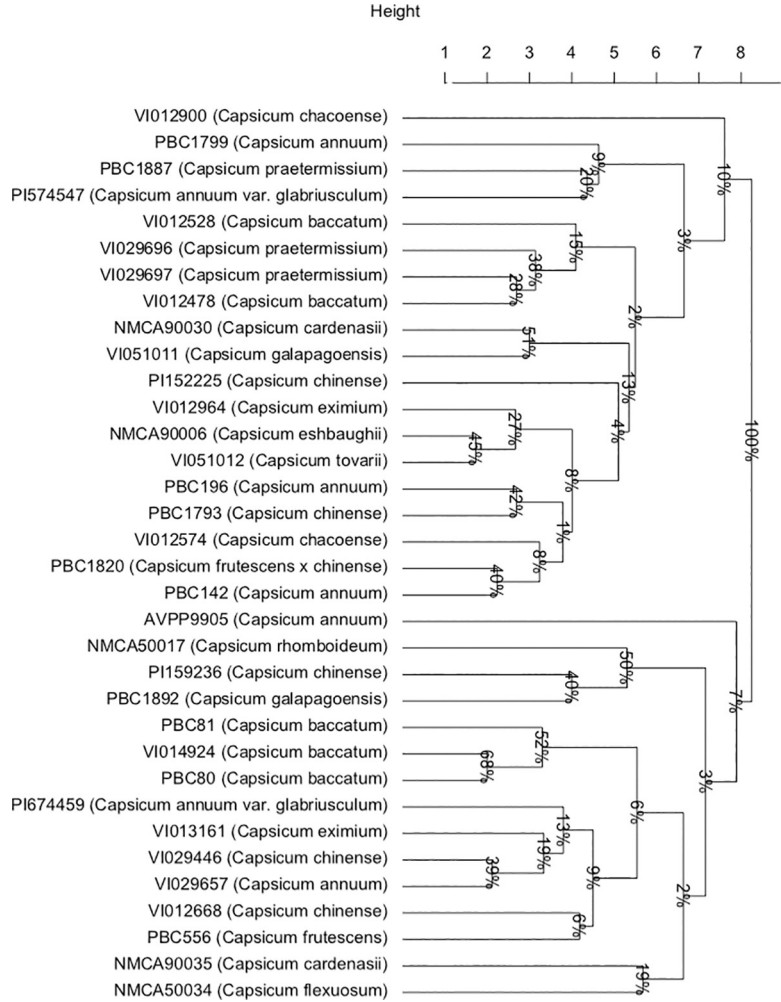

**Fig 6. Unweighted pair group method (UPGMA) clustering of wild and domesticated *Capsicum* species based on the qualitative phenotypic data, scored according to IPGRI descriptors of *Capsicum* scoring method [33].** Bootstrap resampling applied to clusters, represented as percent confidence interval.

59.9% of the total variation. The *C. baccatum* accessions made up a group along with PBC 196 and VI01668, due to their correlated fruit and flower characteristics (pedicel length, fruit width, fruit length, anther length, filament length, corolla length) (Fig 5). *Capsicum annuum* accessions made up a less distinct group, along with the wild progenitor *C. annuum glabriusculum*, and the other domesticated species *C. chinense*, *C. frutescens*, *C. frutescens × chinense* along with *C. eximium* (Fig 5).

The UPGMA clustering of the accessions' qualitative phenotypic data more closely mirrored the genetic relatedness based on SSR molecular markers, especially for the *C. baccatum* and *C. praetermissum* accessions (Fig 6), which formed two groupings (VI012528, VI029696, VI029697; and PBC 80, VI014924, PBC 81). However, we found *C. annuum* did not form a unique clade, highlighting the phenotypic diversity of this domesticated species (Fig 6).

## Discussion

Understanding the relatedness between accessions of *Capsicum* species, and the extent to which they hybridize is key in identifying candidates for the introgression of traits of interest

into commercial varieties. Our results mirror the widely accepted species phylogeny; clustering based on genotyping is centered around *C. annuum*, *C. baccatum* and *C. chinense* complexes [4,6,7]. Similarly, our results of phenotyping the accessions also support previously described genetic complexes [5]. Generally, accessions from the domesticated species (*C. annuum*, *C. chinense*, and *C. frutescens*) were nearer the origin of the score plot, with the exception of members of the domesticated *C. baccatum*, which clustered further away from the other domesticated species (Fig 5). Conversely, members of the wild species were further from the origin, indicating greater diversity (Fig 5). Although we measured different phenotypic traits, our findings contradict those of Luna-Ruiz et al. [45], who found greater levels of diversity among domesticated species for capsaicinoids. Interestingly, based on hybridization success rates, there was a weak relationship between relatedness and crossability, which is in contrast with previous understanding that compatibility between complexes is low [9]. This suggests potential for crop improvement with wild relatives of domesticated species using genetic bridge strategies.

Genotyping using SSR markers evenly distributed across the genome provides evidence of the level of relatedness between wild and domesticated species, and has become a valuable tool for this purpose in many species [46–53]. The use of SSR markers is particularly useful when little is known about the species in question, as in the study of wild species of *Capsicum*, which are relatively poorly understood. We have supplemented genotyping using SSR markers with targeted sequences of the *waxy* gene. The sequence of this single-copy nuclear gene encoding the granule-bound starch synthase (GBSS, also known as *waxy*) protein has been previously utilized in elucidating phylogenies in *Capsicum* [5,54,55], and has proven useful in understanding interspecies relationships here.

When phylogeny, interspecific compatibility and phenotype are considered in concert, the identity of a number of accessions included in this study may be questioned. The issue of misidentification of *Capsicum* species has been raised previously, with several genebanks incorrectly reporting accessions *C. frutescens* as *C. galapagoense* (P.W. Bosland, pers. comm.). Thorough characterization is important in supporting conservation of genetic material and identifying gaps in genebank collections [56]. Only 12% of national vegetable germplasm collections have been characterized biochemically, while 65% have been characterized morphologically [56]. Thorough characterization is therefore key in understanding the reproductive relationships between *Capsicum* species.

Clustering based on SSR genotyping revealed a close relationship between *C. baccatum* accessions (Fig 1), as expected for this well-accepted domesticated species. *Capsicum praetermissium* accessions are also grouped within this complex, shown by both SSR and *waxy* genotyping, which supports previous findings [57] and suggestions that *C. praetermissium* in fact comprises a subgroup of *C. baccatum* [57]. *Capsicum praetermissium* is thought to have diverged prior to domestication of *C. baccatum*, but has not yet been utilized in breeding domestic *C. baccatum* accessions [58]. We found *C. praetermissium* readily hybridized with *C. baccatum* (Fig 3) in line with the findings of Emboden Jr. [6], and thus offers potential as a genetic resource.

*Capsicum chinense* species accessions comprised a significant cluster, which included *C. chinense*, *C. frutescens*, *C. eshbaughii*, and *C. galapagoense* (Fig 1). The grouping of *C. chinense* adjacent to *C. baccatum* was in line with a recent study that also used SSR molecular markers to characterize *Capsicum* species [59]. Conversely, our analysis of publicly available *waxy* sequences found *C. chinense* to be grouped with *C. annuum* var. *glabriusculum*, supporting findings of Pickersgill et al. [14] and Ince et al. [60], who grouped *C. chinense* within the *C. annuum* complex. Furthermore, in this study, a total of 20 crosses were achieved between *C. annuum* (including the wild progenitor *C. annuum* var. *glabriusculum*), and *C. chinense*, 13 of which had a *C. chinense* female parent (Fig 3). Seeds from two of these crosses were sown

(VI029446 × PBC 1969 and PI 152225 × NMCA90030) and germinated well (Fig 4). This contrasts to previous work that reports a barrier to reproduction between *C. annuum* and *C. chinense* [61]. However, Costa et al. [16] found that crosses between *C. chinense* and *C. annuum* accession were possible. These findings highlight the genetic variation that exists in *Capsicum* species, as well as the variability in compatibility, and its dependence on accession selection.

The grouping of *C. frutescens* in the *C. chinense* complex (Fig 1) was in line with previous findings of the close relationship of these species [62]. A number of researchers argued their identities as sister species within the annuum clade [57,63] including Walsh and Hoot [54], who similarly used the *waxy* gene sequence in order to delineate phylogenetic relationships among *Capsicum* species. Furthermore, we found *C. frutescens* hybridized readily with both members of the *C. baccatum* and *C. annuum* clades, as well as with *C. chinense* (Fig 3). Of the three *C. frutescens* hybrids selected for sowing, 80% of the PBC 556 × PBC 1970 hybrid seeds germinated (Fig 4). The relationship of *C. eshbaughii* to this clade, and its pairing with *C. eximium* both in SSR and *waxy* genotyping (Figs 1 and 2) was consistent with its previous placement in the 'Purple Corolla clade' [5]. Furthermore, this was supported by Carrizo Garcia et al [5] and Walsh and Hoot [54], whose use of *waxy* gene sequencing demonstrated *C. eximium* as a divergent species, distinct from *C. annuum*. *Capsicum chinense* formed hybrids with other members of this grouping (Fig 3), and 100% of seeds from the *C. chinense* and *C. eximium* cross germinated (Fig 4).

Interestingly, the *C. eximium* and *C. cardenasii* accessions in our study appeared distantly related (Fig 1). This contradicts the relationship seen between accessions of these species in *waxy* sequencing, and previous reports of these species as members of the *C. pubescens* complex [64,65]. Furthermore, their phenotypes correlated closely with *C. annuum* accessions (Fig 5). This raises the question of the validity of the identification of accessions VI013161 and VI012964 as *C. eximium*.

The *C. annuum* accessions comprise a major grouping adjacent to the *C. baccatum* group (Fig 1). A sample of *C. annuum* accessions (PBC 1799, PBC 196, PBC 1867 and AVPP9905) formed a tightly clustered group, indicating genetic similarity. They also display highly correlated phenotypes, forming a cluster along with accessions from other domesticated species (Fig 5). The *C. annuum* accessions PBC 142 and VI029657 were in an adjacent group (Fig 1), therefore may be considered part of the wider *C. annuum* complex, along with *C. chacoense* accession VI012574, *C. galapagoense*, VI051011, and *C. tovarii* accession VI051012. The presence of *C. chacoense* (VI012574) in this group, distant from the second *C. chacoense* accession (VI012900) included in this study, highlights its possible misidentification. Sequencing clustered VI012574 closely with *C. galapagoense* accession VI051011, which may be considered a member of the *C. annuum* complex (Fig 2). Principal component analysis (Fig 5) revealed VI012574 was grouped with *C. annuum* accessions, away from its counterpart, while UPGMA analysis further highlights this disparity. Direct observation of the phenotypes emphasizes the similarity between the morphology of VI012574 and typical *C. annuum* features. This includes upright growth, elongated fruits, and relatively large flowers with blue anthers.

The *C. galapagoense* accession, PBC 1892 was grouped with the wider *C. baccatum* cluster (Fig 1), conflicting previous findings that *C. galapagoense* is derived from a *C. annuum* progenitor population [66]. No successful hybridizations were achieved between PBC 1892 and PBC 556 (*C. frutescens*), which clustering suggested were closely related. The second *C. galapagoense* accession included in the study, VI051011, was distant from PBC 1892 in *waxy* sequence (Fig 2), and grouped within the *C. annuum* complex in the SSR analysis (Fig 1). It also displayed a distinctly different phenotype to that of PBC 1892; PBC 1892 had a compact growth habit, very small fruits, flowers and leaves, and densely pubescent stems and leaves, typical of *C. galapagoense* descriptions. Conversely, VI051011 had a morphology similar to

that of *C. annuum*, reflected in its close proximity to the PCA origin, along with *C. annuum* accessions. Eight hybridizations were achieved between VI051011 and *C. annuum* accessions, and of the selected hybrid seeds sown, 20% germinated. The close clustering of VI051011 with the *C. annuum* complex, their similar morphology and their ability to hybridize suggests likely misidentification of this accession.

There were five further clusters consisting of *C. chacoense*, *C. cardenasii*, *C. flexuosum*, *C. minutifolium*, and *C. rhomboideum* respectively, which had increasingly distant relation to the three major species complexes (Fig 1). Although *C. chacoense* has been previously grouped within the *C. baccatum* clade [5,64], this wild species has an apparently distant relationship with *C. baccatum*, supported by analysis of *waxy* sequencing. *Capsicum cardenasii* was similarly distantly related to other clades. Other studies [60,64,65] also found *C. chacoense* and *C. cardenasii* not to be closely related to any major clade. Furthermore, VI012900 hybridized readily with members of both *C. annuum* and *C. baccatum* clades (Fig 3). Both *C. chacoense* and *C. cardenasii* accessions (with the exception of PI 159236 and PI 15225) lay on the periphery of the PCA plot, clustering with neither *C. baccatum* or *C. annum* groups. This suggests *C. cardenasii* and *C. chacoense* accessions are not members of either *C. baccatum* or *C. annuum* clades. In their *waxy* sequence analysis, Walsh and hoot [54] similarly demonstrated the distinction of *C. chacoense* from either *C. annuum* or *C. baccatum* groups.

*Capsicum flexuosum*, *C. minutifolium* and *C. rhomboideum* were distantly related to the major clades in analysis of both *waxy* sequence and SSR data (Figs 1 and 2), consistent with the body of literature [5,59,66]. The *C. flexuosum* accession (NMCA50034) was also distinct in phenotype from other accessions (Figs 4 and 5). A small number of hybridizations were achieved between *C. flexuosum* and *C. minutifolium* with members of both *C. annuum* and *C. baccatum* clades. However, no hybridizations were achieved between *C. rhomboideum* and any other accession. This finding is supported by the sequence dissimilarity of the *waxy* gene obtained from NCBI, where *C. rhomboideum* clustered with the tomato outgroup and not the other *Capsicum* species (Fig 2). This low success rate of hybridization with a *C. rhomboideum* parent is likely caused by differences in chromosome number, resulting in abnormal chromosomal pairing and disrupting meiosis; however, more studies are needed to confirm this.

The results reported here highlight the extent of phenotypic diversity in *Capsicum* species, the complexity of *Capsicum* phylogeny, and the similarly complex reproductive relationships between *Capsicum* species. The evidence suggesting the incorrect identification of VI013161, VI012964, VI012574, and VI051011 may highlight a broader issue of misidentification of *Capsicum* in genebanks. Thorough characterization of *Capsicum* genetic material taking a multifaceted approach is therefore important for the development of future breeding programs. Furthermore, the generation of diverse hybrids among accessions of all species included in this study (with the exception of *C. rhomboideum*) demonstrates the possibility for introgression of a diverse range of traits of interest directly or through the design of bridge crossing strategies. Wild relatives of domesticated *Capsicum* species therefore represent significant potential for future breeding programs, and should not be discounted on the basis of their assumed relatedness to domesticated species.

## Acknowledgments

We thank Dr. Paul Bosland of Chile Pepper Institute, New Mexico State University, USA for providing *Capsicum* accessions.

## Author Contributions

**Conceptualization:** Derek W. Barchenger.

**Data curation:** Catherine Parry.

**Funding acquisition:** Derek W. Barchenger.

**Investigation:** Catherine Parry, Yen-Wei Wang, Shih-wen Lin.

**Methodology:** Catherine Parry, Yen-Wei Wang, Shih-wen Lin.

**Project administration:** Derek W. Barchenger.

**Supervision:** Derek W. Barchenger.

**Writing – original draft:** Catherine Parry.

**Writing – review & editing:** Yen-Wei Wang, Shih-wen Lin, Derek W. Barchenger.

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
