## [Decision Letter · Decision Letter 0]

14 Jan 2021

PONE-D-20-37323

Reproductive Compatibility in Capsicum is not Reflected in Genetic or Phenotypic
Similarity Between Species Complexes

PLOS ONE

Dear Dr. Barchenger,

Thank you for submitting your manuscript to PLOS ONE. After careful consideration, we
feel that it has merit but does not fully meet PLOS ONE’s publication criteria as it
currently stands. Therefore, we invite you to submit a revised version of the
manuscript that addresses the points raised during the review process.

Please submit your revised manuscript by Feb 28 2021 11:59PM. If you will need more
time than this to complete your revisions, please reply to this message or contact
the journal office at plosone@plos.org. When
you're ready to submit your revision, log on to https://www.editorialmanager.com/pone/ and select the 'Submissions
Needing Revision' folder to locate your manuscript file.

If you would like to make changes to your financial disclosure, please include your
updated statement in your cover letter. Guidelines for resubmitting your figure
files are available below the reviewer comments at the end of this letter.

We look forward to receiving your revised manuscript.

Kind regards,

Dengcai Liu, PhD

Academic Editor

PLOS ONE

Journal Requirements:

Reviewers' comments:

Reviewer's Responses to Questions

**Comments to the Author**

1. Is the manuscript technically sound, and do the data support the conclusions?

Reviewer #1: Partly

Reviewer #2: Yes

2. Has the statistical analysis been performed
appropriately and rigorously? 

Reviewer #1: N/A

Reviewer #2: Yes

3. Have the authors made all data underlying the
findings in their manuscript fully available?

Reviewer #1: No

Reviewer #2: Yes

4. Is the manuscript presented in an intelligible
fashion and written in standard English?

Reviewer #1: Yes

Reviewer #2: Yes

5. Review Comments to the Author

Reviewer #1: In this study, thirty-eight accessions of 15 species of Capsicum were
chosen for investigating genetic diversity and hybridization compatibility as well
as the relationship between species relatedness and their ability to form hybrids
based on multiple methods including phylogentic reconstruction, phenotypic data, and
artificial hybrid. As the authors correctly pointed out that “interspecies
compatibility is not necessarily reflected in relatedness according to established
Capsicum genepool complexes”. The evolutionary factors involved in the establishment
of polyploids in nature may depend, at least, on the parental origin of particular
genomic features (e.g. high level of genetic heterogeneity) and genetic character
(e.g. the ph gene that control chromosome pairing). Tracing the successful factors
in the establishment of hybrids firstly require a robust and clear phylogenetic
framework and then integrate multiple disciplines to give a power evidence for
clarifying the relationship between interspecies compatibility and relatedness. In
fact, this is lacking in this study.

As outlined in the comments below, moreover, I feel there are some substantial issues
(including rewriting) to address.

1. For the first time in the paper, the appearance of species name including C.
annuum (L.), C. baccatum (L.), C. chinense (Jacq.), C. frutescens (L.), and C.
pubescens, should be given a full taxonomic nomenclature.

2. Many basically biological features of the genus Capsicum involving ploidy level,
geographical distribution, the level of reproductive isolation should be described
briefly in introduction. This would help the reader to understand the process of
speciation, potential species relatedness, possible pre- and post-zygotic barriers
to hybridization, and introgression in the genus

3. One of the objectives of this study was to elucidate the relationship between
interspecies compatibility and relatedness through extensive interspecific
hybridization and the construction of a phylogeny. How can you get it?

4. The same species without different accession number should be listed closely
rather than being scattered in different volume in the table (For example, Capsicum
annuum). The table should also present the species in ploidy level, origin ect.

5. A table should be given to show which sample(s) are included in reciprocal
hybridizations.

6. Why the author uses the SSR for studying the relationship among sampled Capsicum
species. In other word, what’s the advantage of SSR in genetic diversity study?
Similarly, why the author selected the wax gene as a marker for phylogenetic
reconstruction. Generally, single- or low-copy genes are less likely subject to
concerted evolution, thus making themselves ideal tools for studying the origin and
evolution of taxa, especially in hybrid speciation.

In addition, I don’t know how the authors get the wax sequences, by direct sequencing
or clone sequencing?

Moreover, many information in phylogenetic reconstruction is lacking. But, the author
present a phylogenetic tree inferred from the wax sequences. do you use a species as
outgroup?

7. Since the experiments of artificial hybridizations were performed, results of
meiotic pairing behavior should be shortly described to give the readers a clear
figure.

8. I can’t pick up the species and their the groupings in the Figure 1 because the
species information is lacking in figure. This actually lead to some inconvenience
to readers.

9. Some species (e.g. C. eximium) seems to be non-monophyletic origin, but there is
no explanation in discussion.

Reviewer #2: Generally, the research demonstrated that the inter-specific
compatibility may not necessarily reflect in relatedness from established Capsicum
gene pool, and correction of the previous mis-identification of a number of Capsicum
species This information will be valuable for future breeding programs.

The following comments and suggestions may help improve the quality of the
manuscript:

Lane 37: The single sequence repeat (SSR) must be changed as simple sequence repeat
(SSR).

Lane 41-42: The description of “The World … wild relatives” can be deleted.

Lane71-72: The description of “This suggestion between complex is not total” is not
clear, it should be rephrased.

Lane 91: The ability to hybridize might be crossability.

Table 1. The Source might have a table note to refer the abbreviations for the
organizations.

The μl might be changed as μL through the main text.

Lane 287-289: The sentence is not clear and needs to be edited.

The Discussion part, the cluster analysis between Fig.1 and Fig. 5 should be compared
in detail, and the contribution of genome sequencing of Capsicum for the
phylogenetic study may also be mentioned.

6. PLOS authors have the option to publish the peer
review history of their article (what does this mean?). If published, this will
include your full peer review and any attached files.

If you choose “no”, your identity will remain anonymous but your review may still be
made public.

**Do you want your identity to be public for this peer review?** For
information about this choice, including consent withdrawal, please see our
Privacy Policy.

Reviewer #1: No

Reviewer #2: No

---

## [Author Response · Author response to Decision Letter 0]

8 Feb 2021

Reviewer #1: In this study, thirty-eight accessions of 15 species of Capsicum were
chosen for investigating genetic diversity and hybridization compatibility as well
as the relationship between species relatedness and their ability to form hybrids
based on multiple methods including phylogentic reconstruction, phenotypic data, and
artificial hybrid. As the authors correctly pointed out that “interspecies
compatibility is not necessarily reflected in relatedness according to established
Capsicum genepool complexes”. The evolutionary factors involved in the establishment
of polyploids in nature may depend, at least, on the parental origin of particular
genomic features (e.g. high level of genetic heterogeneity) and genetic character
(e.g. the ph gene that control chromosome pairing). Tracing the successful factors
in the establishment of hybrids firstly require a robust and clear phylogenetic
framework and then integrate multiple disciplines to give a power evidence for
clarifying the relationship between interspecies compatibility and relatedness. In
fact, this is lacking in this study.

As outlined in the comments below, moreover, I feel there are some substantial issues
(including rewriting) to address.

1. For the first time in the paper, the appearance of species name including C.
annuum (L.), C. baccatum (L.), C. chinense (Jacq.), C. frutescens (L.), and C.
pubescens, should be given a full taxonomic nomenclature.

The authority has been added to the first mention of each species as well as in Table
1. We did not expand the abbreviated genus after first mention, as it is standard to
use the abbreviated genus name after first mention (per PLOS one style). 

2. Many basically biological features of the genus Capsicum involving ploidy level,
geographical distribution, the level of reproductive isolation should be described
briefly in introduction. This would help the reader to understand the process of
speciation, potential species relatedness, possible pre- and post-zygotic barriers
to hybridization, and introgression in the genus

The introduction has been improved to include a description of the barriers to
hybridization that are currently understood in Capsicum. The ploidy level of
Capsicum species has been included, as well as a brief description of the
geographical origin. 

3. One of the objectives of this study was to elucidate the relationship between
interspecies compatibility and relatedness through extensive interspecific
hybridization and the construction of a phylogeny. How can you get it?

This reviewer comment is not clear. The authors request further clarification on this
point so we can appropriate respond and make necessary changes for the improvement
of our manuscript. One possible response is that we constructed phylogenies based on
SSR markers, targeted sequencing of the waxy gene and based on various phenotypic
traits. The relatedness using these techniques was compared and contrasted with
cross compatibility. 

4. The same species without different accession number should be listed closely
rather than being scattered in different volume in the table (For example, Capsicum
annuum). The table should also present the species in ploidy level, origin ect.

Table 1 was heavily revised- we listed members of the same species together and add
the full species name, including authority, for each species in the table. 

5. A table should be given to show which sample(s) are included in reciprocal
hybridizations.

All were used for crossing; therefore, such a table would not differ from Table 1.
Thus, it is not necessary to make a new table for this. Furthermore, it is already
clearly stated in the body of the materials and methods that “Reciprocal
hybridizations were attempted among all combinations of accessions throughout the
experimental period.” and therefore, no change was made here. 

6. Why the author uses the SSR for studying the relationship among sampled Capsicum
species. In other word, what’s the advantage of SSR in genetic diversity study?
Similarly, why the author selected the wax gene as a marker for phylogenetic
reconstruction. Generally, single- or low-copy genes are less likely subject to
concerted evolution, thus making themselves ideal tools for studying the origin and
evolution of taxa, especially in hybrid speciation. In addition, I don’t know how
the authors get the wax sequences, by direct sequencing or clone sequencing?
Moreover, many information in phylogenetic reconstruction is lacking. But, the
author present a phylogenetic tree inferred from the wax sequences. do you use a
species as outgroup?

The use of SSRs, especially those evenly distributed across the genome as in our
case, are widely used to support relatedness among wild and domesticated species,
especially in situations where very little information is known about some of the
wild species, like in Capsicum. The waxy gene has been previously used to help
clarify the species of Capsicum in the past and we used the published makers of
Carrizo García et al., 2016 as already cited in materials and methods and discussion
points have been added. Nowhere in the manuscript did we state that understanding
origin or evolution was an objective of our study. We want to understand relatedness
and how that impacts cross combability. Therefore, the use of SSRs supplemented by
targeted sequencing of the waxy gene and comparisons made with morphology is
appropriate to answer these questions. We have added the tomato waxy gene as an
outgroup. Furthermore, we have added several waxy gene sequences for different
Capsicum species available on NCBI to our analysis, providing confidence in our
results. 

7. Since the experiments of artificial hybridizations were performed, results of
meiotic pairing behavior should be shortly described to give the readers a clear
figure.

The role of meiotic paring was described in the results and discussion section. 

8. I can’t pick up the species and their the groupings in the Figure 1 because the
species information is lacking in figure. This actually lead to some inconvenience
to readers.

This change was made so that species and accession are listed in each figure. 

9. Some species (e.g. C. eximium) seems to be non-monophyletic origin, but there is
no explanation in discussion.

The reviewers perceived non-monophyletic origin is likely misidentification based on
the waxy gene sequence, the SSR markers, and the phenotypic characters. This is
extensively discussed in the manuscript; therefore, no change was made. 

Reviewer #2: Generally, the research demonstrated that the inter-specific
compatibility may not necessarily reflect in relatedness from established Capsicum
gene pool, and correction of the previous mis-identification of a number of Capsicum
species This information will be valuable for future breeding programs.

The following comments and suggestions may help improve the quality of the
manuscript:

Lane 37: The single sequence repeat (SSR) must be changed as simple sequence repeat
(SSR).

This change was made throughout the manuscript. 

Lane 41-42: The description of “The World … wild relatives” can be deleted.

This sentence provides justification for the need of our study. The crossability of
Capsicum has conflicting reports, therefore, in addition to understanding the
hybridization success rate we also need to ensure that the species listed in
genebank repositories are accurate, which can help resolve the contradictions in
research findings. Therefore, no change was made. 

Lane71-72: The description of “This suggestion between complex is not total” is not
clear, it should be rephrased.

Total was changed to absolute here.

Lane 91: The ability to hybridize might be crossability.

The term “ability to hybridize” is very similar to “crossability”, but the author
understand that hybridization refers to the ability of the plants to produce
successful offspring, while crossability is the study of hybridization. To say two
species are crossable does not have the level of accuracy as to say they hybridize.
No change was made here. 

Table 1. The Source might have a table note to refer the abbreviations for the
organizations.

A footnote was added to Table 1 expanding the abbreviations of the sources. 

The μl might be changed as μL through the main text.

PLOS one requires the use of the universal system of units as prescribed by the
Bureau International des Poids et Mesures, which states that liter is abbreviated
with a lowercase “l”. Linked below is the brochure confirming this. Therefore, no
change was made here. 

https://www.bipm.org/utils/common/pdf/si-brochure/SI-Brochure-9-EN.pdf

Lane 287-289: The sentence is not clear and needs to be edited.

This sentence has been revised to indicate that the wild species were dispersed
further from the origin of the plot. 

The Discussion part, the cluster analysis between Fig.1 and Fig. 5 should be compared
in detail, and the contribution of genome sequencing of Capsicum for the
phylogenetic study may also be mentioned.

We cannot conduct any analysis to compare the data presented in these two figures.
However, further discussion has been added of the contribution of waxy sequencing in
both our study and its context in wider literature.

---

## [Editor Report · Decision Letter 1]

16 Feb 2021

PONE-D-20-37323R1

Reproductive Compatibility in Capsicum is not Reflected in Genetic or Phenotypic
Similarity Between Species Complexes

PLOS ONE

Dear Dr. Barchenger,

Thank you for substantial revisions and submitting your manuscript to PLOS ONE again.
After careful consideration, we feel that it can be accepted after minor revison.
Therefore, we invite you to submit a revised version of the manuscript again.

Please submit your revised manuscript by Apr 02 2021 11:59PM. If you will need more
time than this to complete your revisions, please reply to this message or contact
the journal office at plosone@plos.org. When
you're ready to submit your revision, log on to https://www.editorialmanager.com/pone/ and select the 'Submissions
Needing Revision' folder to locate your manuscript file.

If you would like to make changes to your financial disclosure, please include your
updated statement in your cover letter. Guidelines for resubmitting your figure
files are available below the reviewer comments at the end of this letter.

We look forward to receiving your revised manuscript.

Kind regards,

Dengcai Liu, PhD

Academic Editor

PLOS ONE

Journal Requirements:

Additional Editor Comments (if provided):

Dear Dr. Barchenger,

Thank your great efforts to improvethe manuscript.

I have two minor suggestions:

(1) In the title, I think "....is not reflected..." is so strong. I suggest to revise
the title as such "Reproductive compatibility in Capsicum is not
**necessarily **reflected in genetic or
phenotypic similarity between species complexes", or someone else.

(2) please check the taxonomic nomenclature. For instances in page 52-53,* C.
annuum* (L.),* C. baccatum* (L.), C. chinense (Jacq.), C.
frutescens (L.), and C. pubescens (Ruiz & Pav.) should be* C.
annuum* L., *C. baccatum* L., *C.
chinense* Jacq., *C. frutescens* L., and *C.
pubescens *Ruiz & Pav.

---

## [Author Response · Author response to Decision Letter 1]

16 Feb 2021

(1) In the title, I think "....is not reflected..." is so strong. I suggest to revise
the title as such "Reproductive compatibility in Capsicum is not necessarily
reflected in genetic or phenotypic similarity between species complexes", or someone
else.

This change was made and the title is now Reproductive Compatibility in Capsicum is
not Necessarily Reflected in Genetic or Phenotypic Similarity Between Species
Complexes

(2) please check the taxonomic nomenclature. For instances in page 52-53, C. annuum
(L.), C. baccatum (L.), C. chinense (Jacq.), C. frutescens (L.), and C. pubescens
(Ruiz & Pav.) should be C. annuum L., C. baccatum L., C. chinense Jacq., C.
frutescens L., and C. pubescens Ruiz & Pav.

This change was made throughout the manuscript.

---

## [Editor Report · Decision Letter 2]

18 Feb 2021

Reproductive Compatibility in Capsicum is not Necessarily Reflected in Genetic or
Phenotypic Similarity Between Species Complexes

PONE-D-20-37323R2

Dear Dr. Barchenger,

We’re pleased to inform you that your manuscript has been judged scientifically
suitable for publication and will be formally accepted for publication once it meets
all outstanding technical requirements.

Kind regards,

Dengcai Liu, PhD

Academic Editor

PLOS ONE
---

## [Editor Report · Acceptance letter]

24 Feb 2021

PONE-D-20-37323R2 

Reproductive Compatibility in *Capsicum* is not Necessarily Reflected
in Genetic or Phenotypic Similarity Between Species Complexes 

Dear Dr. Barchenger:

I'm pleased to inform you that your manuscript has been deemed suitable for
publication in PLOS ONE. Congratulations! Your manuscript is now with our production
department. 

Kind regards, 

on behalf of

Dr. Dengcai Liu 

Academic Editor

PLOS ONE